# The Need of Enterococcal Coverage in Severe Intra-Abdominal Infection: Evidence from Animal Study

**DOI:** 10.3390/jcm10051027

**Published:** 2021-03-02

**Authors:** Min Ji Lee, Tae Nyoung Chung, Ye Jin Park, Han A. Reum Lee, Jung Ho Lee, Chang June Yune, Jinkun Bae, Sehwan Mun, Jeong Su Park, Kyuseok Kim

**Affiliations:** 1Department of Emergency Medicine, CHA University School of Medicine, Seongnam 13497, Korea; minji.lee29@gmail.com (M.J.L.); hendrix74@cha.ac.kr (T.N.C.); yejin6577@naver.com (Y.J.P.); harlee91@naver.com (H.A.R.L.); ning3135@naver.com (J.H.L.); june1976@hanmail.net (C.J.Y.); galen97@chamc.co.kr (J.B.); bogmi0415@gmail.com (S.M.); 2Department of Laboratory Medicine, Seoul National University Bundang Hospital, Seongnam 13620, Korea; mdmicrobe@gmail.com

**Keywords:** sepsis, intraabdominal infection, antibiotics, mortality

## Abstract

Intra-abdominal infection (IAI) is a common and important cause of infectious mortality in intensive care units. Adequate source control and appropriate antimicrobial regimens are key in the management of IAI. In community-acquired IAI, guidelines recommend the use of different antimicrobial regimens according to severity. However, the evidence for this is weak. We investigated the effect of enterococcal coverage in antimicrobial regimens in a severe polymicrobial IAI model. We investigated the effects of imipenem/cilastatin (IMP) and ceftriaxone with metronidazole (CTX + M) in a rat model of severe IAI. We observed the survival rate and bacterial clearance rate. We identified the bacteria in blood culture. We measured lactate, alanine aminotransferase (ALT), creatinine, interleukin (IL)-6, IL-10, and reactive oxygen species (ROS) in the blood. Endotoxin tolerance of peripheral blood mononuclear cells (PBMCs) was also estimated to determine the level of immune suppression. In the severe IAI model, IMP improved survival and bacterial clearance compared to CTX + M. Enterococcus spp. were more frequently isolated in the CTX + M group. IMP also decreased plasma lactate, cytokine, and ROS levels. ALT and creatinine levels were lower in IMP group. In the mild-to-moderate IAI model, however, there was no survival difference between the groups. Immune suppression of PBMCs was observed in IAI model, and it was more prominent in the severe IAI model. Compared to CTX + M, IMP improved the outcome of rats in severe IAI model.

## 1. Introduction

Intra-abdominal infection (IAI) is a common condition and the second most frequent cause of infectious mortality in intensive care units [1,2]. Adequate source control and antimicrobial regimens are key in the management of IAI [3].

Guidelines on the management of IAI from North America emphasize the use of appropriate antimicrobials depending on the conditions of the patients, for example, community- or health care-associated IAI, clinical factors predicting failure of source control, etc. Patients with community-acquired IAI are divided into high- or mild-to-moderate-risk groups and different empiric antimicrobial therapies are recommended [3,4]. High-risk patients are recommended carbapenem (imipenem-cilastatin, meropenem, doripenem, etc.) or piperacillin-tazobactam. In contrast, ceftriaxone or cefotaxime with metronidazole is recommended for mild-to-moderate-risk patients. This approach is based on the assumption that high-risk patients have a narrow time window for survival; therefore, broad-spectrum antimicrobials should be administered as initial therapy [3]. However, this hypothesis has not been tested in clinical trials or even in preclinical settings, and it is recommended merely as an expert opinion [3].

Considering the importance of antimicrobials in the management of IAI, it may be important to provide supporting evidence for antimicrobial guidelines for IAI. We hypothesized that the effects of antimicrobials differ according to the severity of IAI. To test this hypothesis, we used a polymicrobial IAI model with different severity. We compared the effects of imipenem–cilastatin (IMP) with those of ceftriaxone with metronidazole (CTX + M) in a polymicrobial IAI models with different severity.

## 2. Materials and Methods

### 2.1. In Vivo Sepsis Model Induction

This study was approved by the Institutional Animal Care and Use Committee of our institute in accordance with the National Institutes of Health Guidelines. Sprague–Dawley rats weighing 270–330 g were used. The rats were housed in a controlled environment with access to standard food and water ad libitum for 7 days before the experiment.

We used a body weight-adjusted polymicrobial IAI sepsis model according to a previous study [5]. In brief, feces were collected from the donor rats. The collected feces were diluted with 5% dextrose saline at a ratio of 1:3. This fecal slurry was vortexed to obtain a homogeneous suspension before administration into the intraperitoneal cavity. The volume of fecal slurry administered to each animal was adjusted based on the body weight of the recipient rat. We then administered subcutaneous fluid resuscitation (30 mL/kg 5% dextrose saline).

Thereafter, the rats were randomly assigned to the IMP and CTX + M groups. The rats in the IMP were injected with IMP subcutaneously at a dose of 25 mg/kg every 12 h per day. The rats in the CTX + M group were administered ceftriaxone (150 mg/kg) once per day and metronidazole (7.5 mg/kg) every 12 h per day. The sham group did not get fecal slurry. In total, 131 rats were used in the study.

Experiment 1: Survival study—In this study, we introduced sepsis at different severities (for severe IAI, we used 5.5 mL/kg fecal slurry, and for mild to moderate IAI, 2.5 mL/kg fecal slurry). In each experiment, IMP, CTX + M, or no antibiotics were administered. Survival was monitored every 12 h for 14 days.

Experiment 2: Tissue study—In this study, 5.5 mL/kg fecal slurry was administered, and blood and tissues were harvested 24 h after sepsis induction. The colony-forming units (CFUs) in the blood and spleen were counted. Bacterial isolation was performed using blood culture. We measured lactate, alanine aminotransferase (ALT), creatinine, interleukin (IL)-6, IL-10, and reactive oxygen species (ROS) in the blood.

Experiment 3: Endotoxin tolerance study—In this study, response of peripheral blood mononuclear cells (PBMCs) to lipopolysaccharide (LPS) stimulation was investigated using TNF-α response to LPS. To determine the differences in response to LPS according to the severity of sepsis, we compared the responses in the mild IAI and severe IAI groups.

### 2.2. Colony Forming Unit (CFU) Assay

Blood and spleen samples were used to count bacterial CFUs in the IAI rat model. Thirty milligrams of spleen tissue was homogenized in 500 μL of Dulbecco’s Phosphate Buffer Saline (DPBS) (Welgene Inc., Gyeongsan, Korea) and centrifuged at 12,000 rpm for 10 min at 4 °C. The cell pellets were resuspended in 1 mL of DPBS (to form stock samples). The stock samples were diluted 1:10 with DPBS. To count the CFUs, 800 μL of DPBS was added to 200 μL of blood. The samples were spread on Tryptic soy agar (TSA) plate (BD bioscience, NJ, USA) without antibiotics and incubated at 37 °C overnight, and the bacterial colonies were counted for analysis.

### 2.3. Bacterial Isolation

Two milliliters of blood from the abdominal aortas of each rat were inoculated into the BD BACTEC Peds Plus/F Culture Vials (Bacton, Dickinson and Company, Shannon, Ireland). After 72 h of incubation at 37 °C, microbiological analysis was performed.

### 2.4. Lactate Assay

Lactate concentrations in the plasma were evaluated using a Lactate Colorimetric Assay Kit (BioVision, Milpitas, CA, USA). Briefly, the sample and reaction mix buffer were added to the wells and incubated for 30 min at room temperature. After 30 min, the optical density of each well was measured at 450 nm using a microplate reader (VersaMax with SoftMax Pro software, Molecular Devices, CA, USA).

### 2.5. ALT and Creatinine

Serum ALT level was measured using an alanine aminotransferase (ALT or SGPT) activity colorimetric/fluorometric assay kit (#K752-100, BioVison, CA, USA). The optical density at 570 nm was measured using a microplate reader (VersaMax with SoftMax Pro software, Molecular Devices, CA, USA).

Serum creatinine levels were measured using a creatinine colorimetric/fluorometric assay kit (#K625-100, BioVison, CA, USA). The optical density at 570 nm was measured using a microplate reader (VersaMax with SoftMax Pro software, Molecular Devices, CA, USA).

### 2.6. Cytokine Measurements

The plasma levels of the cytokines IL-6 (R6000B, R&D Systems, MN, USA) and IL-10 (ab214566, Abcam, MA, USA) were measured using Enzyme-linked immunosorbent assay (ELISA) kits according to the manufacturer’s instructions. The optical density at 450 nm was measured using a microplate reader (VersaMax with SoftMax Pro software, Molecular Devices, CA, USA).

### 2.7. Assessment of Plasma Levels of Reactive Oxygen Species (ROS)

Plasma DCF levels were analyzed according to the manufacturer’s instructions as previously described. [6] In brief, DCF was measured using H2DCFDA (Invitrogen, Carlsbad, CA, USA).

### 2.8. Ex vivo Peripheral Blood Mononuclear Cell (PBMC) Stimulation with LPS

PBMCs were isolated using the Ficoll gradient method, as previously described [7]. Isolated PBMCs were stimulated with LPS to observe and compare the level of immune paralysis. TNF-α levels were measured 5 h after LPS stimulation. In brief, isolated PBMCs were seeded at a density of 1 × 105 cells/mL in 96-well plates, and 100 ng/mL LPS (Escherichia coli O111: B4; Sigma-Aldrich, St. Louis, MO, USA) was added to each well. After 5 h, the culture medium was collected, and TNF-α level was analyzed using a TNF-α ELISA kit (ab236712, Abcam, MA, USA).

### 2.9. Statistical Analysis

The Shapiro–Wilk test was performed to determine the normality of the data. Normally distributed data were presented as mean ± standard deviation and compared using independent *t*-tests. If the data did not fit a normal distribution, they were presented as the median and interquartile range and were analyzed using the Mann–Whitney U test or Kruskal–Wallis test. Fisher’s exact test was performed for categorical variables. Survival rates according to the type of antibiotics were compared using Kaplan–Meier survival analysis with the log-rank test. A *p*-value of <0.05 was defined as statistically significant. All analyses were performed with the SigmaPlot Version 14.5 (SigmaPlot, IL, USA).

## 3. Results

### 3.1. Effects of Different Antibiotics in the Rat Intra-abdominal Infection (IAI) Model

#### 3.1.1. Survival Rate and Plasma Lactate Levels

In the severe IAI model, the survival rate significantly increased with IMP (78% vs. 7.6% for 14 days) (Figure 1A). In the mild-to-moderate IAI model, however, there was no significant change in the survival rate between the treatment groups (Figure 1B).

#### 3.1.2. Bacterial Isolation and CFU

CFUs had significantly decreased in the IMP group in the severe IAI model (Figure 2). In the blood culture, only E. gallinarum was isolated, and the positive culture rates were significantly different between the groups (Table 1). In the severe IAI model, the CTX + M group showed 100% (6/6) positivity, while 50% (3/6) positivity was observed in the IMP group. In the mild-to-moderate model, the CTX + M group showed higher positive results than the IMP group (66.7% vs. 0%). In total, the CTX + M group showed significantly high rates of positive blood cultures for E. gallinarum (83.3% vs. 25.0%, *p* = 0.0123). Severe IAI groups had a higher positive rate (75.0% vs. 33.3%, *p* = 0.0995) than the non-severe IAI groups.

#### 3.1.3. Lactate, Alanine Aminotransferase (ALT), and Creatinine

Plasma lactate level and serum ALT, and serum creatinine levels were significantly decreased in IMP group in the severe IAI model (Figure 3). IAI induced lactic acidosis, and treatment of IMP attenuated it compared to CTX + M (2.7 ± 0.6 vs. 2.0 ± 0.3 nmol/microliter). IAI also induced liver and kidney injury, and it was mitigated in IMP group compared to CTX + M group (serum ALT; 103.2 ± 31.3 vs. 66.6 ± 20.6 IU/L, serum creatinine; 2.7 ± 0.5 vs. 2.0 ± 0.3 nmol/microliter).

#### 3.1.4. Cytokines

Plasma IL-6 and IL-10 levels were significantly reduced in the IMP group under the severe IAI model (Figure 4) (IL-6; 1273.9 ± 560.3 vs. 848.8 ± 427.1 pg/mL, IL-10; 590.1 ± 248.2 vs. 296.2 ± 101.1 pg/mL).

#### 3.1.5. Reactive Oxygen Species (ROS)

IMP significantly reduced the DCF-DA levels in the severe IAI group (Figure 4) (median 5185, IQR 6655 vs median 2531, IQR 1183 arbitrary unit).

### 3.2. Immune Suppression Effect of IAI Model

TNF-α production was significantly reduced when the PBMCs of the IAI model were stimulated with LPS, and this was significantly prominent in the severe IAI model, indicating increased immune suppression in the severe IAI sepsis model (Figure 5).

## 4. Discussion

This is the first study to demonstrate the different effects of IMP and CTX + M in a polymicrobial IAI model. Clinical outcomes such as survival rate, bacterial clearance, and ALT, creatinine, and lactate levels were in favor of IMP therapy in the severe IAI model (survival rate 78% vs. 7.6% for 14 days). However, these differences were not present in the mild-to-moderate IAI model (mortality rate 0% in 14 days).

Current guidelines on IAI recommend risk assessment for treatment failure and death [3,4]. They also recommend different antibiotic strategies according to the risk of death or severity of infection. For example, for high-risk patients or severe infections, carbapenem or piperacillin-tazobactam is recommended. On the contrary, third-generation ceftriaxone or cefotaxime with metronidazole is the usual recommendation for low-risk IAI. However, the evidence for this is weak, and the recommendations are based on the assumption that patients with high risk or severe infection have a narrow therapeutic window, so the initial use of broad-spectrum antibiotics is reasonable. In this study, we demonstrated that IMP is more effective than CTX + M in severe IAI. In mild-to-moderate IAI, however, IMP and CPX showed the same efficacy, supporting current guidelines [3,4]. However, this should be interpreted with caution because this model mimics community acquired IAI instead of hospital associated IAI.

Previous studies on IAI have usually identified polymicrobial infections, and E. coli, Bacteroides, Streptococcus spp., and Enterococcus spp. have been isolated [8,9,10]. In this study, only E. gallinarum was isolated. Enterococcus sp. is a common inhabitant of the human gastrointestinal tract and is also a frequent opportunistic pathogen, so immunocompromised patients are vulnerable to enterococcal infections [11,12]. Although Enterococcus faecium and E. faecalis are most common enterococcal infections in human, in certain situations, such as in immunocompromised hosts, E. gallinarum has been shown as the cause of severe infections, including bacteremia, endocarditis, meningitis, and hospital-acquired infections [13,14,15,16,17,18]. We used healthy and young animals, so underlying immunosuppression can be excluded. However, E. gallinarum was isolated in the blood culture, which might mean that our IAI sepsis model led to immunosuppressed status. It is well known that sepsis-induced immunosuppression could develop after severe primary infection or trauma [19,20]. This study also showed the immunosuppression of PBMCs, evidenced by a reduced response to LPS stimulation (endotoxin tolerance), especially in the severe IAI sepsis model. Endotoxin tolerance is considered one of the mechanisms of immune suppression in sepsis patients [21,22,23,24].

The CTX + M group showed higher positive blood culture rates for E. gallinarum culture than the IMP group. The IMP group showed more favorable outcomes such as survival and acute liver/kidney injury than the CTX + M group under the severe IAI model. IMP covers Enterococcus spp., but CTX + M does not, which might explain the results of this study. However, we found that all animals survived in the mild IAI model regardless of the antibiotics used, indicating natural clearance of Enterococcus spp. in the mild IAI model. Thus, Enterococcus spp. could be clinically important in severe IAI, wherein severe immune suppression may occur. The results of this study support this concept because the severe IAI model showed more immune suppression in terms of endotoxin tolerance than the mild IAI model. The mortality from enterococcal bacteremia is known to be high, especially in immunocompromised patients [18]. This indicates the importance of antibiotic coverage of Enterococcus spp. in immunocompromised IAI patients. Taken together, we can infer that the severe IAI, but not mild-to-moderate IAI, might cause severe immunosuppression, which leads Enterococcus spp. to be pathogenic and clinically meaningful, which should be covered with appropriate antibiotics.

High plasma cytokine levels are known to be associated with high mortality [25,26], and IMP altered cytokine profiles in favor of survival and organ preservation compared to the CTX + M group.

This study has several limitations. First, we did not perform source control, which is important and routinely performed in clinical practice. It is not easy to perform this procedure in small animals, and this could produce another bias. Moreover, the conditions were the same in each group, and without source control, animals under the mild-to-moderate IAI model showed 100% survival. Second, we observed the immune suppression of PBMCs, but we could not specify the cell types. PBMCs include T and B lymphocytes, NK cells, monocytes, dendritic cells, and macrophages, and we could not determine which immune cells were responsible for the immunosuppressive effects of IAI-induced sepsis. However, it is well known that lymphocytes, monocytes, and macrophages are responsible for sepsis-induced immunosuppression [27]. Lastly, we used fecal slurry IAI model to simulate IAI in clinical situation. However, the intrinsic limitations of this model could exist. For example, in a more natural kind of IAI not all bowel bacteria may translocate in the same way, and to the same amount as fecal slurry IAI model.

## 5. Conclusions

IMP with enterococcal coverage improved the outcome of the groups in the severe IAI model as compared to CTX + M. However, this benefit was not seen in the groups under the mild-to-moderate IAI. This difference could be due to the severity of sepsis. In severe sepsis, more immune-suppression could occur, which caused opportunistic infection such as enterococcal infection, which needs antimicrobial coverage. This experiment supports current guidelines for IAI.

## Figures and Tables

**Figure 1 jcm-10-01027-f001:**
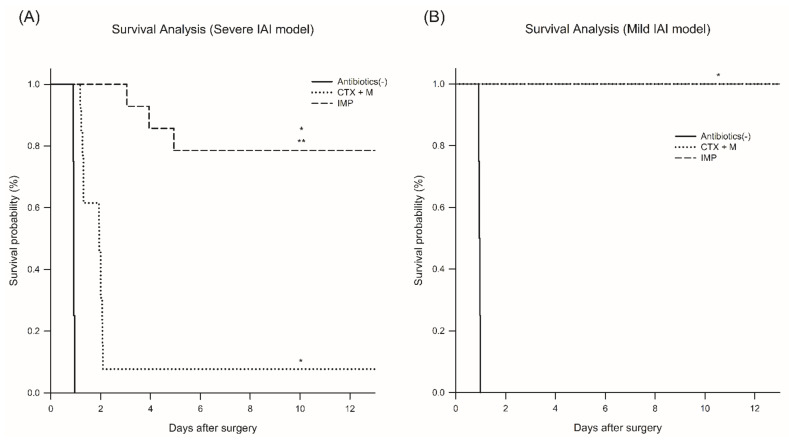
Survival effect of different antibiotic treatments on polymicrobial intra-abdominal infection (IAI) model. (**A**) Severe IAI model, *n* = 4 for no antibiotics, *n* = 13 for CTX + M, *n* = 14 for IMP. (**B**) Mild IAI model, *n* = 4 per group, * *p* < 0.05 compared with no antibiotics model, ** *p* compared with CTX + M model. IAI, intraabdominal infection; CTX + M, ceftriaxone with metronidazole; IMP, imipenem.

**Figure 2 jcm-10-01027-f002:**
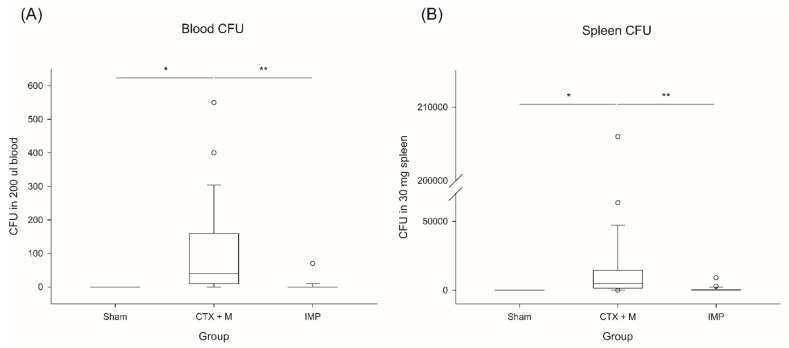
CFU in blood and spleen in severe IAI model. (**A**) CFU in blood, (**B**) CFU in spleen. * *p* < 0.05 compared with sham, ** *p* compared with CTX + M model. IAI, intraabdominal infection; CTX + M, ceftriaxone with metronidazole; IMP, imipenem; CFU, colony-forming unit.

**Figure 3 jcm-10-01027-f003:**
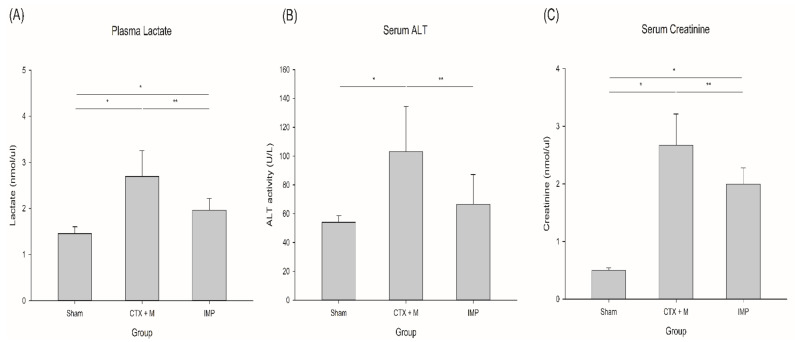
Effects of different antibiotic treatments on organ injury in severe IAI model. (**A**) Plasma lactate, (**B**) serum ALT, (**C**) serum creatinine. *n* = 3 for sham, *n* = 14 for IMP and CTX + M, **p* < 0.05 compared with sham, ** *p* compared with CTX + M model. IAI, intraabdominal infection; CTX + M, ceftriaxone with metronidazole; IMP, imipenem; ALT, alanine aminotransferase.

**Figure 4 jcm-10-01027-f004:**
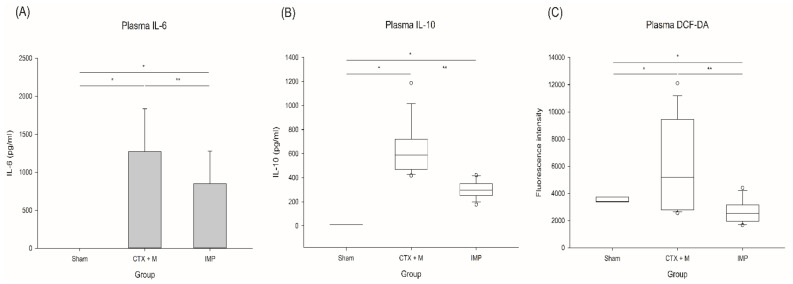
Inflammation and oxidative stress in severe IAI model. (**A**) IL-6, (**B**) IL-10, (**C**) ROS generation. *n* = 3 for sham, *n* = 14 for IMP and CTX + M, * *p* < 0.05 compared with sham, ** *p* compared with CTX + M model. IAI, intraabdominal infection; CTX + M, ceftriaxone with metronidazole; IMP, imipenem; ROS, reactive oxygen species; DCF-DA, 20,70-dichlorofluorescein diacetate.

**Figure 5 jcm-10-01027-f005:**
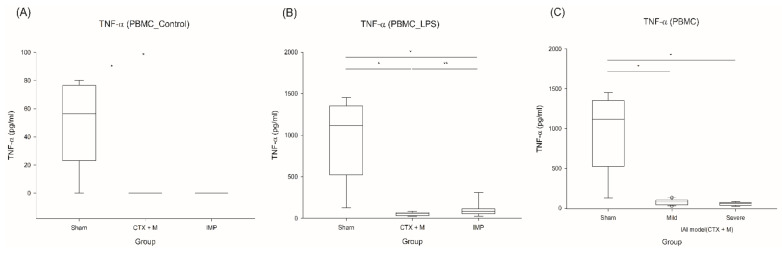
Effects of different antibiotics on endotoxin tolerance of PBMC stimulated by LPS. (**A**) Baseline TNF-α in PBMCs after severe IAI induction, (**B**) endotoxin tolerance of PBMCs in severe IAI model, (**C**) endotoxin tolerance of PBMCs in mild or severe IAI model with CTX + M. *n* = 5 for sham, *n* = 6–11 for other groups; * *p* < 0.05 compared with sham, ** *p* compared with CTX + M model. IAI, intraabdominal infection; CTX + M, ceftriaxone with metronidazole; IMP, imipenem; PBMC, peripheral blood mononuclear cell; LPS, lipopolysaccharide.

**Table 1 jcm-10-01027-t001:** Blood Cultures of Rats.

	2.5 CTX + M	2.5 IMP	5.5 CTX + M	5.5 IMP	*p*-value
**Enterococcus (-)**	2	6	0	3	0.00482
**Enterococcus (+)**	4	0	6	3	

Blood cultures were obtained 24h after sepsis model induction.

## Data Availability

Data are the property of the authors and can become available by contacting the corresponding author.

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
