# Peer review of "The Need of Enterococcal Coverage in Severe Intra-Abdominal Infection: Evidence from Animal Study"

_jcm, 2021, doi:10.3390/jcm10051027_

Round 1

Reviewer 1 Report

The manuscript describes a rat animal model to simulate mild to severe intraabdominal infections (IAI). The authors assessed animal survival rates after setting an IAI and administration of two Abx regimes. Abx treatment was chosen according to existing guidelines. The authors counted CFU in animal organs and assessed immunologic responses (IL, ROS, etc.). I am a microbiologist and enterococcal expert and less a specialist for IAI and animal models; so, my comments rather refer to more microbiological aspects of the given study.

Introduction

The authors should specify for which region of the world the recommendations for IAI they are discussing, are valid/considered. Differences could be expected between Europe, North America and Asia.

Figures 2 -5. Please explain abbreviation “Sham”. If Sham should be the control group of infected animals that did not receive antibiotics I don’t understand the figures. Shouldn’t be infected animals without receiving antibiotics at all show highly elevated CFU counts in body organs (Figure 2), elevated marker levels (Figure 3), elevated IL level (Fug. 4) ? Or is Sham the group of uninfected animals?

Chapters 3.1.3. to 3.1.6. A little more data than a single sentence would be advantageous. For instance, if the level of … is significantly increased; please explain, increased to what.

Limitations

I see some more limitations. Please add and discuss them. (1) The kind of setting an IAI infection seems a bit artificial: in a more natural kind of IAI not all bowel bacteria may translocate in the same way and to the same amount as exemplified here. Please discuss this aspect. (2) E. gallinarum may be a natural component of a rat (ruminant) and bird intestinal microbiomes; however, it is not for humans. IAI caused by or with the contribution of E. gallinarum in humans are unusual (unknown?) and thus results of this study cannot be simply transferred to the enterococcal components in IAI in humans. The latter may concentrate around E. faecalis or E. faecium. Please discuss this important aspect as well.

Conclusions

Please increase the content of the Conclusions by one or two more sentences. To my mind, this is a bit too short. I miss a line break after the Conclusions and before the “Author contributions” part.

Author Response

The manuscript describes a rat animal model to simulate mild to severe intraabdominal infections (IAI). The authors assessed animal survival rates after setting an IAI and administration of two Abx regimes. Abx treatment was chosen according to existing guidelines. The authors counted CFU in animal organs and assessed immunologic responses (IL, ROS, etc.). I am a microbiologist and enterococcal expert and less a specialist for IAI and animal models; so, my comments rather refer to more microbiological aspects of the given study.

Introduction

The authors should specify for which region of the world the recommendations for IAI they are discussing, are valid/considered. Differences could be expected between Europe, North America and Asia.

àWe appreciate your helpful comments. We discussed guidelines from IDSA and Surgical infection society, both of which represent North America. We added this in introduction as follows; Guidelines on the management of IAI from North America emphasize the use of appropriate antimicrobials depending on the conditions of the patients

Figures 2 -5. Please explain abbreviation “Sham”. If Sham should be the control group of infected animals that did not receive antibiotics I don’t understand the figures. Shouldn’t be infected animals without receiving antibiotics at all show highly elevated CFU counts in body organs (Figure 2), elevated marker levels (Figure 3), elevated IL level (Fug. 4) ? Or is Sham the group of uninfected animals?

àWe apologize for the confusion. Sham group did not get fecal slurry. We added this one to the method section as follows; The sham group did not get fecal slurry. 

Chapters 3.1.3. to 3.1.6. A little more data than a single sentence would be advantageous. For instance, if the level of … is significantly increased; please explain, increased to what.

 àAs your recommendations, we added little more data on each chapters.

Limitations

I see some more limitations. Please add and discuss them. (1) The kind of setting an IAI infection seems a bit artificial: in a more natural kind of IAI not all bowel bacteria may translocate in the same way and to the same amount as exemplified here. Please discuss this aspect.

à I appreciate your comments. As your recommendations, we added limitations as follows; Lastly, we used fecal slurry IAI model to simulate IAI in clinical situation. However, the intrinsic limitations of this model could exist. For example, in a more natural kind of IAI not all bowel bacteria may translocate in the same way, and to the same amount as fecal slurry IAI model.

(2) E. gallinarum may be a natural component of a rat (ruminant) and bird intestinal microbiomes; however, it is not for humans. IAI caused by or with the contribution of E. gallinarum in humans are unusual (unknown?) and thus results of this study cannot be simply transferred to the enterococcal components in IAI in humans. The latter may concentrate around E. faecalis or E. faecium. Please discuss this important aspect as well.

  • We appreciate your important comments. The following literature showed that although Enterococcus faecium, E. faecalis are most common enterococcal infections in human, in certain situations, such as in immunocompromised hosts, E. gallinarum has been shown as the cause of severe infections, including bacteremia, endocarditis, meningitis, and hospital-acquired infections. Therefore, we added this one in discussion as follows; Although Enterococcus faecium and E. faecalis are most common enterococcal infections in human, in certain situations, such as in immunocompromised hosts, E. gallinarum has been shown as the cause of severe infections, including bacteremia, endocarditis, meningitis, and hospital-acquired infections.
  •  
  • References:
  • Characterization of an Enterococcus gallinarum Isolate Carrying a Dual vanA and vanB Cassette, J Clin Microbiol. 2015 Jul;53(7):2225-9.

Enterococcus gallinarum endocarditis occurring on native heart valves. J Clin Microbiol. 2002;40:2308–2310.

First report of vanA Enterococcus gallinarum dissemination within an intensive care unit in Argentina. Int J Antimicrob Agents. 2005; 25:51–56.

Clinical and epidemiological features of Enterococcus casseliflavus/flavescens and Enterococcus gallinarum bacteremia: a report of 20 cases. Clin Infect Dis. 2001;32:1540–1546.

Nosocomial outbreak of Enterococcus gallinarum: untaming of rare species of enterococci. J Hosp Infect. 2008; 70:346–352.

Bacteremia caused by non-faecalis and non-faecium Enterococcus species at a medical center in Taiwan, 2000 to 2008. J Infect. 2010;61:34–43.

Conclusions

Please increase the content of the Conclusions by one or two more sentences. To my mind, this is a bit too short. I miss a line break after the Conclusions and before the “Author contributions” part.

  • We appreciate your advices. We added more contents to the conclusion as follows; IMP with enterococcal coverage improved the outcome of the groups in the severe IAI model as compared to CTX+M. However, this benefit was not seen in the groups under the mild-to-moderate IAI. This difference could be due to the severity of sepsis. In severe sepsis, more immune-suppression could occur, which caused opportunistic infection such as enterococcal infection, which needs antimicrobial coverage. This experiment supports current guidelines for IAI.

Reviewer 2 Report

Min Ji Lee et al. in this paper, aimed to assess the effects of imipenem/cilastatin (IMP) and ceftriaxone with metronidazole (CTX + M) in a rat model of severe Intra-abdominal infection (IAI).

This is a in vivo sepsis model induction on the rats that were ramdomly assigned to receive imipenem/cilastatin or ceftriaxone with metronidazole.

The development of sepsis is evoked by injection of feces into the intraperitoneal cavity.

It is a very interesting study also in consideration of the increasingly evident resistance to antibiotics.

In general, the study is well designed but some improvements have to be performed.

-The authors should stress the argument that this intra-abdominal sepsis is comparable to a community-acquired intra-abdominal infection and not to a nosocomial infection.

- The authors could add in the introduction and discussion, in the light of the results obtained, a possible difference in the outcome and response to antibiotic therapy based on microbiological isolations and related resistance spectra

-The authors can specify on how many rats have carried out the study

Author Response

Min Ji Lee et al. in this paper, aimed to assess the effects of imipenem/cilastatin (IMP) and ceftriaxone with metronidazole (CTX + M) in a rat model of severe Intra-abdominal infection (IAI).

This is a in vivo sepsis model induction on the rats that were ramdomly assigned to receive imipenem/cilastatin or ceftriaxone with metronidazole.

The development of sepsis is evoked by injection of feces into the intraperitoneal cavity.

It is a very interesting study also in consideration of the increasingly evident resistance to antibiotics.

In general, the study is well designed but some improvements have to be performed.

 -The authors should stress the argument that this intra-abdominal sepsis is comparable to a community-acquired intra-abdominal infection and not to a nosocomial infection.

àWe appreciate very important comment. We added this to discussion as follows: However, this should be interpreted with caution because this model mimics community acquired IAI instead of hospital associated IAI.

- The authors could add in the introduction and discussion, in the light of the results obtained, a possible difference in the outcome and response to antibiotic therapy based on microbiological isolations and related resistance spectra

àWe appreciate your comment. We discussed it in first draft as follows; The CTX+M group showed higher positive blood culture rates for E. gallinarum cul-ture than the IMP group. The IMP group showed more favorable outcomes such as sur-vival and acute liver/kidney injury than the CTX+M group under the severe IAI model. IMP covers Enterococcus spp., but CTX+M does not, which might explain the results of this study

-The authors can specify on how many rats have carried out the study  

àWe appreciate your comments. In each figure, there are numbers of rats used in experiment.

However, there is no total number of rats used. Therefore, we added this to the method section as follows: In total, 131 rats were used in this study.